# A Novel Classification Method Using the Takagi–Sugeno Model and a Type-2 Fuzzy Rule Induction Approach

Martin Tabakov *, Adrian B. Chlopowiec and Adam R. Chlopowiec

Department of Artificial Intelligence, Wroclaw University of Science and Technology, 50-370 Wroclaw, Poland; 254517@student.pwr.edu.pl (A.B.C.); 254518@student.pwr.edu.pl (A.R.C.)
* Correspondence: martin.tabakow@pwr.edu.pl

**Abstract:** The main purpose of this research was to introduce a classification method, which combines a rule induction procedure with the Takagi–Sugeno inference model. This proposal is a continuation of our previous research, in which a classification process based on interval type-2 fuzzy rule induction was introduced. The research goal was to verify if the Mamdani fuzzy inference used in our previous research could be replaced with the first-order Takagi–Sugeno inference system. In the both cases to induce fuzzy rules, a new concept of a fuzzy information system was defined in order to deal with interval type-2 fuzzy sets. Additionally, the introduced rule induction assumes an optimization procedure concerning the footprint of uncertainty of the considered type-2 fuzzy sets. A key point in the concept proposed is the generalization of the fuzzy information systems' attribute information to handle uncertainty, which occurs in real data. For experimental purposes, the classification method was tested on different classification benchmark data and very promising results were achieved. For the data sets: Breast Cancer Data, Breast Cancer Wisconsin, Data Banknote Authentication, HTRU 2 and Ionosphere, the following F-scores were achieved, respectively: 97.6%, 96%, 100%, 87.8%, and 89.4%. The results proved the possibility of applying the Takagi–Sugeno model in the classification concept. The model parameters were optimized using an evolutionary strategy.

**Keywords:** fuzzy sets; interval type-2 fuzzy sets; Takagi–Sugeno model; fuzzy information systems; classification; fuzzification optimization

## 1. Introduction

Classification is one of the most important and challenging machine learning tasks. As the main goal is to investigate similarities between groups of objects, the classification accuracy strongly depends on the initial data sets used for training. There are a lot of issues to be handled, such as unbalanced or biased training data, but also incomplete or vague information which is often the case for real data. A lot of research was done to address this problem, especially using fuzzy classification and by extending to type-2 fuzzy sets as well. For example, recently in [1], a new fuzzy reasoning method for an interval type-2 fuzzy classification system including cluster-based rules was introduced. Authors propose to incorporate the introduced reasoning procedure with a new Possibility-based fuzzy measure to handle uncertainty of cluster centers in an interval type-2 fuzzy rule-based classification system. In [2], a robust sparse representation for classification of medical images is proposed based on an introduced adaptive type-2 fuzzy dictionary learning. Remote-sensing image classification techniques using type-2 fuzzy sets were introduced [3–5] where, for example recently, considering the last cited research, a novel robust interval type-2 possibilistic fuzzy clustering model for the classification of complex remote sensing land cover was presented.

There are many fuzzy control applications based on the use of type-2 fuzzy sets. Recently, type-2 fuzzy logic system for ergonomic control of indoor environments was proposed [6]. The system calculates the effective working time and time-dependent change

in carbon dioxide levels and aims to evaluate ergonomic comfort conditions. The authors proved better system performance using type-2 fuzzy sets for ergonomic control in fuzzy environments. Important medical application was proposed in [7]. An interval type-2 fuzzy stochastic modeling and control strategy to consider the uncertainties of the COVID-19 pandemic in order to control the number of infected people was introduced. A common strategy for fuzzy model parameter tuning is the use of evolutionary algorithms. In [8], authors applied a genetic algorithm to tune the parameters of an interval type-2 fuzzy proportional–derivative controller, in order to track the trajectory of a snake robot in the presence of system uncertainties. In [9,10], slime mold and particle swarm optimization algorithms were used for parameter tuning of interval type-2 fuzzy controllers, respectively. In [11], the design of membership functions for interval type-2 fuzzy tracking controllers was optimized using a metaheuristic algorithm.

In general, type-2 fuzzy sets provide better generalizations [12,13] as the main assumption here is the additional fuzzification of the membership values of type-1 fuzzy sets involved. Therefore, type-2 fuzzy sets can handle better with imprecise information [14,15].

Additionally, to deal with information issues, data discovery is used to induce knowledge. Information systems and rough sets, originally introduced by Pawlak [16–18], are often used to represent knowledge. Basic mathematical concepts, using rough sets theory to induce fuzzy rules, have been worked on for a long time. In [19], fuzzy learning methods for the automatic generation of membership functions and fuzzy if-then rules from training examples were introduced. Fuzzy decision rules induction was presented in [20] using the tolerance-based rough sets model. Rough sets were used to reduce the dimensionality of complex datasets as preprocessing for fuzzy rule induction [21].

The above guided us to investigate the possible combination of fuzzy reasoning with Pawlak's information systems. We have already introduced our concept of a fuzzy information system [22]. This research is a continuation of our previous research, by evolving and combining our concept with the Takagi–Sugeno fuzzy model [23]. Fuzzy information systems introduce a relation between attributes and fuzzy sets [24]. Applications of fuzzy information systems can be found in the fields of decision-making [25,26] and rule extraction [26,27]. Recently, new decision-making concepts were introduced in [28] by combining regret theory with three-way decisions in fuzzy incomplete information systems, or in [29], by multi-attribute predictive analysis using fuzzy rough sets.

In this research, a fuzzy information system to induce rules was applied with respect to classification attribute values. Next, a transformation into fuzzy rules was proposed, and finally, the fuzzy sets used were expanded into type-2 fuzzy sets. To provide classification, the Takagi–Sugeno model was applied in order to simplify our previous concept, which was related to the Mamdani fuzzy model [30]. The Takagi–Sugeno model does not require rule consequents defined as fuzzy sets, but introduces additional parameters. For that reason, in this research, an optimization procedure was applied to determine the best fuzzy classifier for the benchmark data considered. The research experiments were provided for binary classification problems defined with well-known benchmark data.

Therefore, our research goal was to provide a classification method, by replacing the Mamdani model used in our previous research with the first-order Takagi–Sugeno fuzzy inference. This simplified the induction of type-2 fuzzy rules in the classification process, as fuzzy sets were not required to be defined as rule consequents. The main research advantage of this work states the proposal of a fuzzy information system used to induce type-2 fuzzy rules, incorporated with the Takagi–Sugeno inference to provide classification.

The rest of the paper is organized as follows: in Section 2, the benchmark data used and some theoretical background are introduced; in Section 3, the introduced classification procedure is explained; in Section 4, the classification results are presented; and finally, Sections 5 and 6 draw corresponding discussion, possible further developments and conclusions.

## 2. Material and Methods

### 2.1. Materials

In this research, classification experiments were provided on well-known benchmark data [31] for binary classification. This is the same benchmark data as those used in our previous study [22], with details given in Table 1.

**Table 1.** The benchmark data used.

| Benchmark | Number of Attributes | Number of Samples | Class Proportions |
|---|---|---|---|
| Blood Transfusion | 7 | 748 | 76%/24% |
| Breast Cancer Data | 4 | 569 | 63%/37% |
| Breast Cancer Wisconsin | 9 | 683 | 65%/35% |
| Data Banknote Authentication | 4 | 1372 | 56%/44% |
| Haberman | 3 | 305 | 73%/27% |
| Heart | 13 | 303 | 46%/54% |
| HTRU 2 | 8 | 17,898 | 9%/91% |
| Immunotherapy | 7 | 90 | 79%/21% |
| Indian Liver Patient | 10 | 583 | 71%/29% |
| Ionosphere | 34 | 351 | 64%/36% |
| Parkinson | 22 | 187 | 74%/26% |
| Pima Indians Diabetes | 8 | 768 | 65%/35% |
| Vertebral | 6 | 310 | 68%/32% |

### 2.2. Methods

#### 2.2.1. Type-1 and Type-2 Fuzzy Sets

A Type-1 fuzzy set F consists of a non-empty domain X and a function $\mu_F : X \to [0, 1]$, called a membership function [32]. Considering a continuous membership function, a fuzzy set of type-1 as defined in Equation (1):

$$F =_{df} \int_X \mu_F(x)/x, x \in X \tag{1}$$

The above integral denotes the collection of all points $x \in X$ with associated membership grade $\mu_F(x) \in [0, 1]$. The function values define the grade of membership of the elements of X to the fuzzy set F. Membership functions are assumed to describe imprecise or vague information.

The type-1 fuzzy sets concept was expanded by fuzzifying the membership function values themselves and type-2 fuzzy sets were introduced. A type-2 fuzzy set, denoted as $\tilde{F}$, is defined in Equation (2) [5]:

$$\tilde{F} =_{df} \int_{x \in X} \int_{u \in J_x} \mu_{\tilde{F}}(x, u)/(x, u), J_x \subseteq [0, 1] \tag{2}$$

where $\iint$ denotes the union over all admissible x and u. The most widely used special cases of type-2 fuzzy sets, mostly because of their easy interpretation, are the interval type-2

(IT2) fuzzy sets [5]. Uncertainty about $\tilde{F}$ is handled by the so-called footprint of uncertainty (FOU) of $\tilde{F}$, as shown in Equation (3):

$$\text{FOU}(\tilde{F}) =_{df} \bigcup_{x \in X} J_x, \; J_x \subseteq [0, 1] \tag{3}$$

The area of a FOU is directly related to the uncertainty that is conveyed by an interval type-2 fuzzy set, and what follows, a FOU with more area is more uncertain than the one with less area. The lower membership function (LMF) and upper membership function (UMF) of $\tilde{F}$ are two type-1 membership functions $\underline{F}$ and $\overline{F}$ that bound the FOU, which are used to describe $J_x$, see Equation (4):

$$J_x =_{df} \left[ \mu_{\underline{F}}(x), \mu_{\overline{F}}(x) \right] \tag{4}$$

which leads to Equation (5):

$$\text{FOU}(\tilde{F}) =_{df} \bigcup_{x \in X} \left[ \mu_{\underline{F}}(x), \mu_{\overline{F}}(x) \right] \tag{5}$$

Figure 1 illustrates the graphical interpretation of the above definitions.

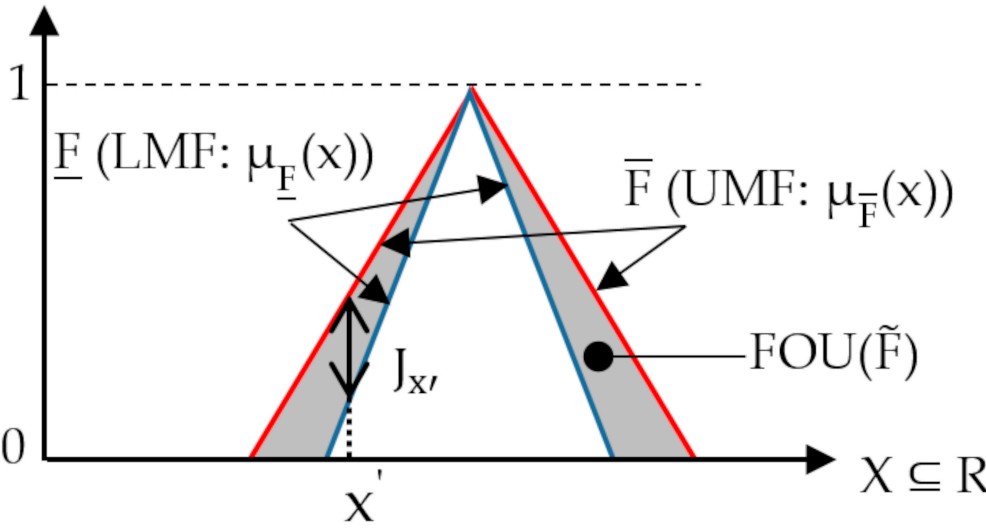

**Figure 1.** An interval type-2 fuzzy set $\tilde{F}$ determined by membership functions of the type-1 fuzzy sets: $\underline{F}$ and $\overline{F}$.

2.2.2. Takagi–Sugeno Type-2 Fuzzy Inference

The major difference between the type-1 and type-2 Takagi–Sugeno fuzzy models obviously lies in the incorporation of type-2 fuzzy sets in the inference process. This requires corresponding fuzzy rule interpretation and type–reducer mechanism.

A type-1 fuzzy rule in terms of the Takagi–Sugeno model, takes the following form:

$$R :_{df} \text{ IF } (x_1 \text{ is } X_1) o \ldots o (x_i \text{ is } X_i) \text{ THEN } f(x_1, \ldots x_i)$$

where $X_i$ ($i = 1, \ldots, I$) are type-1 fuzzy sets defined over corresponding domains. The operator 'o' is assumed as a $\otimes, \oplus : [0, 1]^2 \rightarrow [0, 1]$ are the t- and s-norms [33], respectively. These binary operators are applied in fuzzy logic as generalizations of the conjunction and disjunction Boolean logic operators. The Zadeh's t- and s- norms defined by the min and max operators, were applied, respectively. The rule consequents are multivariable

functions, defined as combinations of polynomials. The most popular solution, i.e., first-order Takagi–Sugeno model, assumes combinations of linear functions. Therefore, any rule consequent takes the form:

Consequent: $f(x_1, \ldots, x_i) =_{df} \sum_{k=1}^{i} (a_k x_k + b_k) = \sum_{k=1}^{i} a_k x_k + \sum_{k=1}^{i} b_k$, where $a_k$ and $b_k$ are the parameters of the corresponding linear functions, defined with respect to each $x_i$.

It is interesting to note that in the Takagi–Sugeno model, there is no need to define the consequents as fuzzy sets and therefore, there is no need to define any membership function in the consequents as it should be done using, for example, the Mamdani model [30].

The final output of the system is the weighted over all rule outputs, as defined in Equation (6):

$$\text{Final Output} =_{df} \frac{\sum_{k=1}^{N} w_k f_k}{\sum_{k=1}^{N} w_k}, \tag{6}$$

where N is the number of rules, $w_k$ is the rule-firing value derived from the $k^{th}$ rule antecedent and $f_k$ is the value of the kth rule consequent.

Considering type-2 fuzzy sets, for an input vector $\bar{x} = \{x_1', x_2', \ldots, x_i'\}$, typical computations of an IT2 fuzzy system consist of the following steps:

(1) Compute the membership intervals of $x_i'$ for each $\tilde{X}_i^n$, $\left[ \mu_{\underline{X}_i^n}(x_i'), \mu_{\overline{X}_i^n}(x_i') \right]$, $i = 1, \ldots, I$; $n = 1, \ldots, N$ (N is the number of rules),

(2) Compute the firing interval of the $n^{th}$ rule:

$$F^n\left(\bar{x}\right) =_{df} \left[ \mu_{\underline{X}_1^n}(x_1') \, o \ldots o \, \mu_{\underline{X}_i^n}(x_i'), \, \mu_{\overline{X}_1^n}(x_1') \, o \ldots o \, \mu_{\overline{X}_i^n}(x_i') \right],$$

(3) Use type reduction to combine $F^n\left(\bar{x}\right)$ with corresponding rule consequents.

The most popular one is the center-of-set (COS) type reducer [34] using the Karnik–Mendel algorithm [34,35] or their modifications [36,37].

In order to apply the Karnik–Mendel algorithm, a single type-2 fuzzy set must be derived using the rules outputs. For a Takagi–Sugeno type-2 fuzzy system, the aggregate set is generated using the following procedure:

1. Sort the rule outputs from all rules into ascending order,
2. For each output, define the UMF value using the maximum firing range of the considered rule,
3. For each output, define the LMF value using the minimum firing range of the considered rule.

As an example, consider the data shown in Table 2. Let us assume five rules with output values and firing range limits. For these data, the aggregate type-2 fuzzy set shown in Figure 2 will be generated.

**Table 2.** Example aggregate set generation.

| Rule | Rule Output Value ($f_k$) | Minimum Firing Value | Maximum Firing Value |
|------|---------------------------|----------------------|----------------------|
| 1 | 6.8 | 0.1 | 0.6 |
| 2 | 1.6 | 0.3 | 0.7 |
| 3 | 5.3 | 0.2 | 0.5 |
| 4 | 0.4 | 0.4 | 0.8 |
| 5 | 3.1 | 0.2 | 0.9 |
| Sorted in ascending order: | | | |
| 4 | 0.4 | 0.4 | 0.8 |
| 2 | 1.6 | 0.3 | 0.7 |

**Table 2.** *Cont.*

| Rule | Rule Output Value ($f_k$) | Minimum Firing Value | Maximum Firing Value |
|------|---------------------------|----------------------|----------------------|
| **5** | **3.1** | **0.2** | **0.9** |
| 3 | 5.3 | 0.2 | 0.5 |
| 1 | 6.8 | 0.1 | 0.6 |

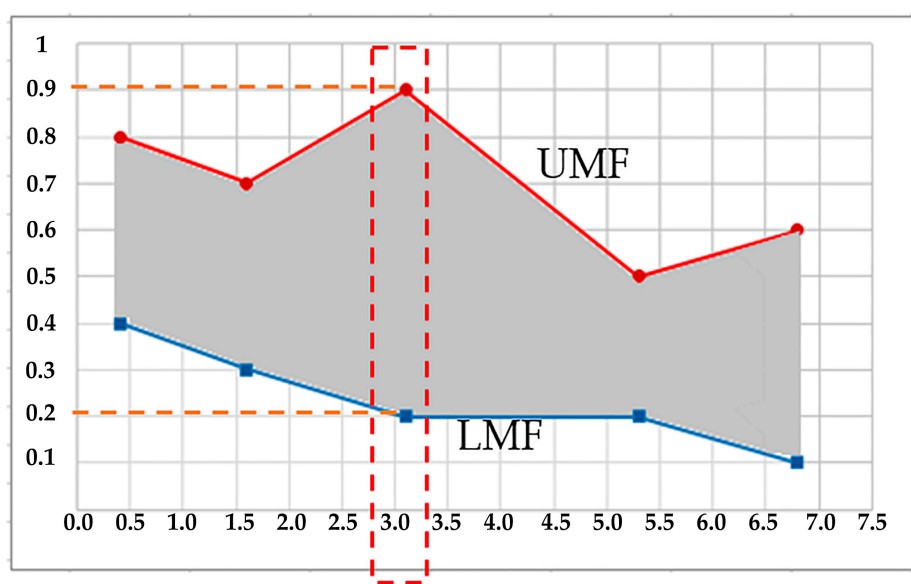

**Figure 2.** Exemplary output type-2 fuzzy set for an input vector. To make it easier to understand the structure of the aggregate set, the values for Rule 5 are marked.

Defining the above aggregate type-2 fuzzy output set, the Karnik–Mendel algorithm [34,35] can be applied in order to derive the final output system value. As the complete Takagi–Sugeno type-2 fuzzy inference procedure is well-described in the literature, we will omit further details.

## 3. Fuzzy Information System with Tagaki-Sugeno Reasoning

### 3.1. Type-1 Fuzzy Information System and Fuzzy Rule Induction

This research is based on our interpretation of a fuzzy information system, introduced in [22]. We assume that the values of the system information function are linguistic variables which corresponds to fuzzy sets. This is a granulation information proposal. A classic information system [16] is defined by the elements: (U, A, V, f), where U is a universe, A is a set of attributes, V represents attributes domains: V = $_{df}$ $\bigcup_a V_a$, with nonempty domain $V_a$ of the a-th attribute (a $\in$ A), and f is the information function f: U×A → V, $\forall x \in U$, a $\in$ A f(u, a) $\in V_a$. The indiscernibility binary relation (IND), defined over U (IND $\subseteq U^2$), plays in theory an important role. It is an equivalence relation, defined in Equation (7):

$$\text{IND(B)} =_{df} \{(u_i, u_j) \in U \text{ ' } U: \forall a \in B \text{ } f(u_i, a) = f(u_j, a)\}, u_i, u_j \in U, B \subseteq A \tag{7}$$

The above binary relation is used to define the lower and upper approximations of any subset of U. Any such a pair is defined as a rough set [17,18].

In accordance with our interpretation of a fuzzy information system over any attribute domain $V_a$, three fuzzy subsets are defined: {low, medium, high}. The information function values are replaced with corresponding fuzzy sets, with respect to the maximum membership value: max{$\mu_{low}$(object, attribute), $\mu_{medium}$(f(object, attribute), $\mu_{high}$(f(object,

attribute)}. The membership function of the medium fuzzy set assumes Gaussian distribution of the attribute data for any a ∈ A.

Therefore, the membership functions for the considered fuzzy sets can be easily defined, see Equation (8):

$$\mu_{\text{medium}}(a) =_{df} e^{\frac{-(a-a_0)^2}{2\sigma^2}}, \ \mu_{\text{low}}(a) =_{df} \left\{ \begin{array}{c} 1 - e^{\frac{-(a-a_0)^2}{2\sigma^2}} \ : \ a \ < a_0 \\ 0 : a \geq \ a_0 \end{array} \right\},$$

$$\mu_{\text{high}}(a) =_{df} \left\{ \begin{array}{c} 0 : a < \ a_0 \\ 1 - e^{\frac{-(a-a_0)^2}{2\sigma^2}} \ : a \geq a_0 \end{array} \right\}, \ a \ \in \ V_a \text{with expected value}(a_0)$$

and standard deviation $(\delta)$

(8)

The above attribute fuzzification introduces a very simple initialization of the information function as well as values' generalization. For example, let us consider the below matrix (Table 3) which represents a fuzzy information system: U = {object$_1$, object$_2$, object$_3$, object$_4$}, A = {attibute$_1$, attribute$_2$, attribute$_3$}, V = V$_{\text{attribute}_1}$ ∪ V$_{\text{attribute}_2}$ ∪ V$_{\text{attribute}_3}$ and we assume for each set V$_{\text{attribute}_i}$ a Gaussian distribution in order to obtain the fuzzy sets {low$_i$, medium$_i$, high$_i$}.

**Table 3.** Fuzzy information system example.

|  | *attribute$_1$* | *attribute$_2$* | *attribute$_3$* |
|---|---|---|---|
| *object$_1$* | *low* | *medium* | *low* |
| *object$_2$* | *medium* | *high* | *high* |
| *object$_3$* | *medium* | *high* | *high* |
| *object$_4$* | *low* | *medium* | *low* |

So, for example, f(object$_2$, attribute$_2$) is introduced as *high* because the following inequality is satisfied: $\mu_{\text{high}}$(f(object$_2$, attribute$_2$)) ≥ max{ $\mu_{\text{low}}$(f(object$_2$, attribute$_2$), $\mu_{\text{medium}}$(f(object$_2$, attribute$_2$)} or f(object$_1$, attribute$_3$) is introduced as *low*, as $\mu_{\text{low}}$(f(object$_1$, attribute$_3$)) ≥ max{ $\mu_{\text{medium}}$(f(object$_1$, attribute$_3$), $\mu_{\text{high}}$(f(object$_1$, attribute$_3$)}.

The corresponding partition (P) with respect to IND has two equivalence classes:

*P*/IND{attribute1, attribute2, attribute3} = {{object1, object4}, {object2, object3}}

Next, if a decision attribute (A*) is added to the set of attributes: A = A ∪ A* an information system can be represented by a decision table. Assuming such an extension, the Skowron and Suraj rule induction method can be applied [38,39] and next, transformation of the rules induced for each decision into fuzzy rules is introduced.

For example, for pairs: (object$_1$, attribute$_1$): low, (object$_2$, attribute$_2$): high, (object$_3$, attribute$_1$): medium, and a rule for decision D is induced as follows:

decision D: (f(object$_1$, attribute$_1$) ∧ f(object$_2$, attribute$_2$)) ∨ f(object$_3$, attribute$_1$),
the above rule can be transformed into the following fuzzy rule:
If ((f(object$_1$, attribute$_1$) is low) ⊗ (f(object$_2$, attribute$_2$) is high)) ⊕ (f(object$_3$, attribute$_1$) is medium) Then D.

An explanation of the rule induction procedure and the generation of fuzzy rules directly from a decision table can be found in our previous research in more detail ([22], Sections 2.2.3 and 2.2.4).

The most important issues here are: (1) the possibility of attribute value generalization by applying fuzzy sets directly as values in the decision table. Therefore, the information function values are related directly to membership functions of fuzzy sets. (2) The rule induction procedure and the transformation of the rule induced into type-1 fuzzy rules.

### 3.2. Involving Type-2 Fuzzy Sets

The idea of applying type-2 fuzzy sets is very simple and related to our previous research [22]. It is enough to change the values of the standard deviation of the Gaussian-type membership function of the fuzzy set medium to define the bounds of the FOU for the type-2 fuzzy set $\widetilde{medium}$. In such a way, the assumed fuzzy sets {low, medium, high} can be replaced with the {$\widetilde{low}$, $\widetilde{medium}$, $\widetilde{high}$} type-2 fuzzy sets. Figure 3 clarifies the issue. This transformation is done after rule induction and transformation into type-1 fuzzy rules.

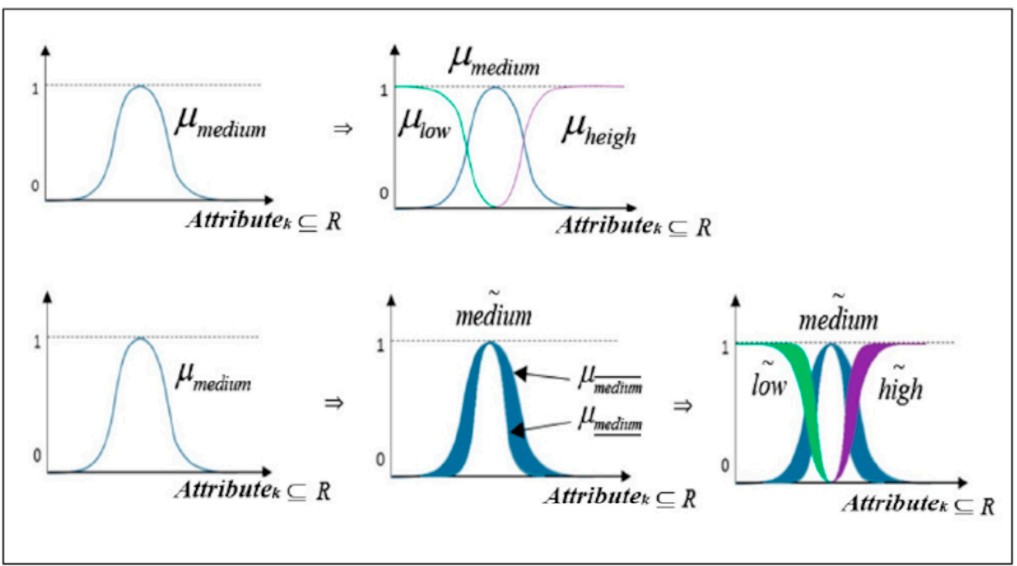

**Figure 3.** The type-1 and type-2 fuzzification process.

Nevertheless, the assumption of Gaussian distribution for each attribute value is hard to be fulfilled. That is the reason why we assumed the experiment with the fuzzy rule premises in terms of standard derivation values (defining the corresponding FOU) parameter named as *sigma_offset* and different configurations of membership functions. Meaning, we extend the assumption of using only three type-2 fuzzy sets {$\widetilde{low}$, $\widetilde{medium}$, $\widetilde{high}$}. Without changing our previous research, the configurations assumed are shown in Table 4 below. This concerns the number of Gaussians in the rule premises, the applied *sigma_offset,* and the expected value for the medium membership function. For more details, see ([22], Section 2.2.6) concerning the fuzzy sets defined in the fuzzy rule premises.

**Table 4.** The IT2 fuzzy sets used (presentation of the concept with respect to a chosen attribute of the blood transfusion dataset, assuming normalization as well).

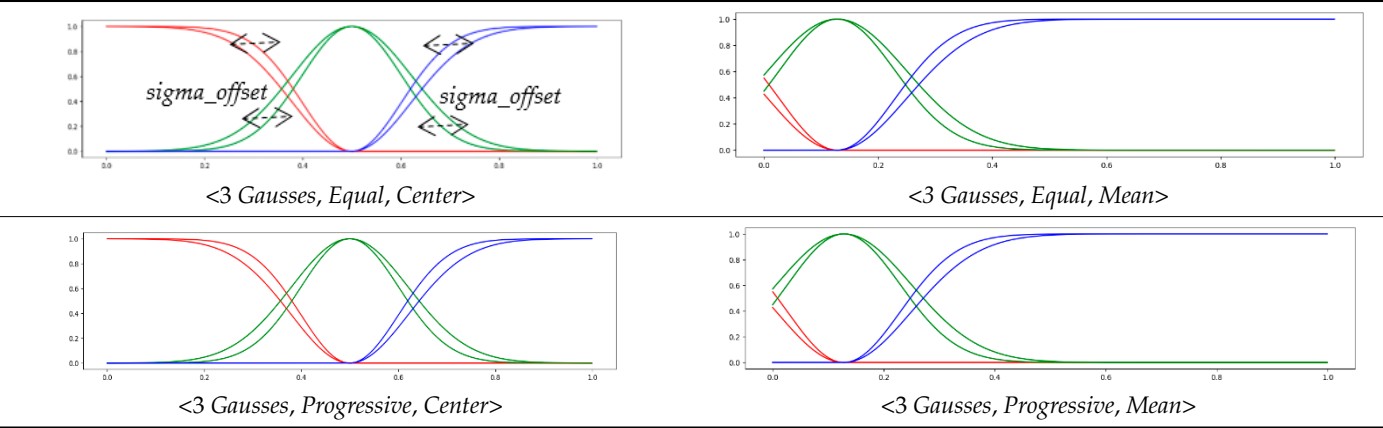

**Table 4.** *Cont.*

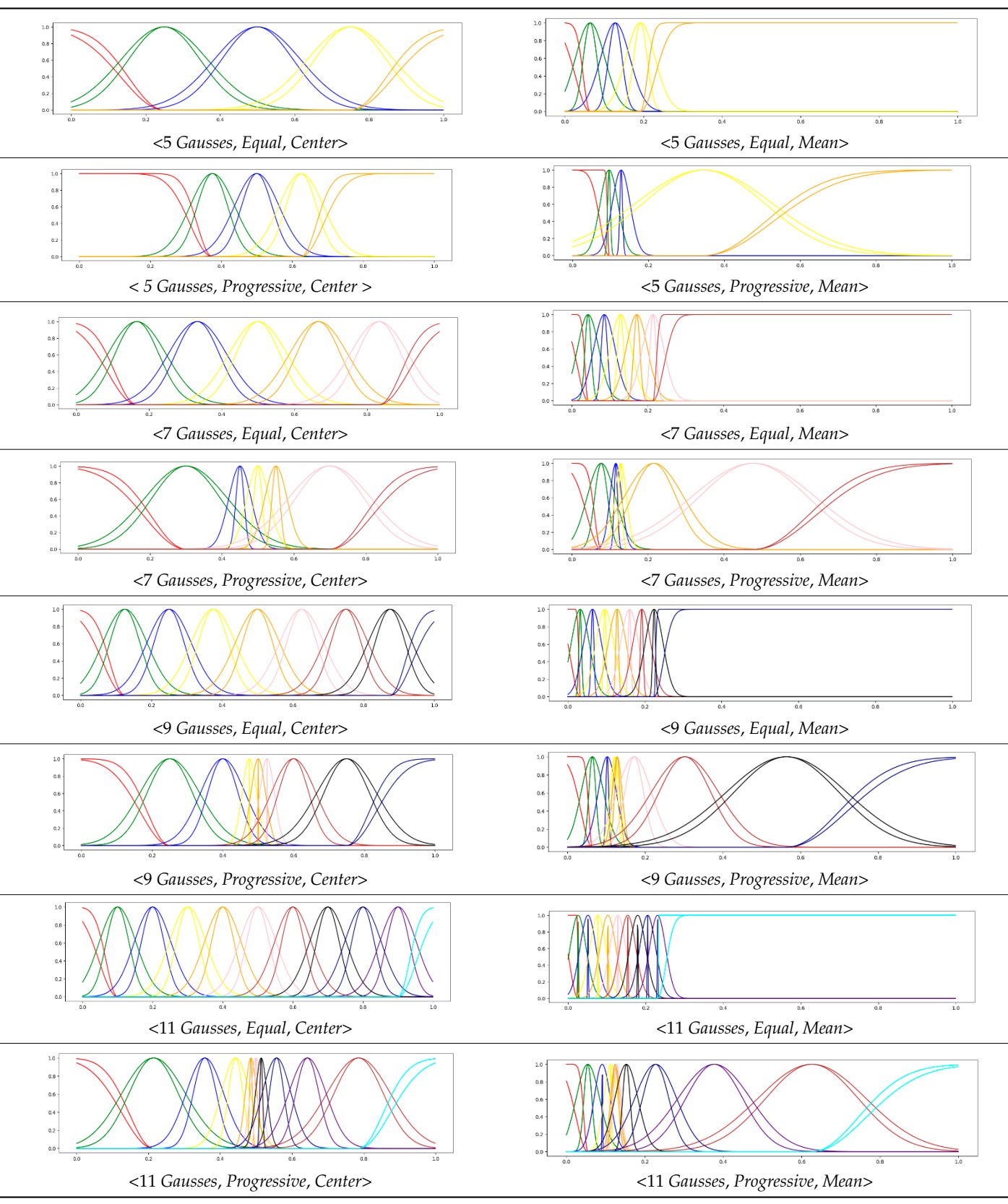

### 3.3. Takagi–Sugeno Reasoning with Optimization

Consequently, for each rule premises, configuration of different *sigma_offset* are considered. Additionally, in terms of first-order Takagi–Sugeno systems, two parameters defining the corresponding linear function for each fuzzy set in the rule premises should be calculated. The assumed varieties of type-2 fuzzy sets along with the *sigma_offset* value and the linear function parameters define an optimization problem. A finite set of membership functions is considered, generated with respect to the *medium* membership function. The assumed numbers of membership functions are: {3, 5, 7, 9 11}. Considering the membership function configurations and the *sigma_offset*, the grid search algorithm in a k-fold cross-validation manner for these hyperparameters optimization was applied. The linear function parameters for each attribute, were optimized using the Covariance Matrix Adaptation Evolutionary Strategy (CMA-ES) [40], i.e., as each cross-validation step defines a new training set, the induced rules also differ. Therefore, at each cross-validation step, we applied CMA-ES for linear function parameters optimization.

Additionally, in the conducted experiments as possible data pre-processing, a dimension reduction procedure using the well-known principal component analysis method (PCA) and the Random Oversampling (ROS) [41–43] to handle with class imbalance were applied. The oversampling was important in terms of rule induction, as the inconsistency elimination step ([22], Section 2.2.3, algorithm 1) strongly favors classes with large presence of data.

The complete set of hyperparameters to be optimized is given in Table 5. On the other hand, Table 6 presents the hyperparameter values used in the CMA-ES procedure itself. The number of folds for each dataset are given in Table 7.

**Table 5.** Tuned hyperparameters of sets of membership functions.

| Hyperparameter | Value |
| --- | --- |
| Number of Gaussian functions | 3, 5, 7, 9, 11 |
| Whether the std applied to Gaussians are the same | Yes/No |
| If the mean value of the medium membership function is derived directly from mean value of the corresponding feature | Yes/No |
| Sigma_offset | [0.5, 0.9] with step of 0.05 |
| Use PCA | Yes/No |
| Use ROS | Yes/No |

**Table 6.** CMA-ES hyperparameters.

| Hyperparameter | Value |
| --- | --- |
| Initialization | Random |
| Maximum evaluations of fitness function | 20,000 |
| Number of restarts | 1 |
| Population size increase ratio after restart | 2 |
| Standard deviation in each coordinate | 0.7 |
| Population size | 20 |
| Stagnation tolerance | 100 evaluations |
| Parameters values range | [−400, 400] |

**Table 7.** Number of folds applied in the cross-validation process.

| Benchmark | k-Value |
| --- | --- |
| Blood Transfusion | 10 |
| Breast Cancer Data | 10 |
| Breast Cancer Wisconsin | 10 |
| Data Banknote Authentication | 10 |
| Haberman | 6 |
| Heart | 5 |
| HTRU | 10 |
| Immunotherapy | 4 |
| Indian Liver Patient | 10 |
| Ionosphere | 6 |
| Parkinson | 4 |
| Pima Indians Diabetes | 10 |
| Vertebral | 5 |

The experimental procedure pipeline in this research was defined as follows:

1.  Split a considered dataset into k-folds.
2.  For each set of hyperparameters:
    a.  For each cross-validation step:
        i.  Set one fold as held-out for validation, use the rest for training.
        ii.  Induce the knowledge base using the training set.
        iii.  Infer the crisp value for each sample in the training set with the Takagi–Sugeno model.
        iv.  Classify the sample with the threshold function set at 0. Treat the sample as negative if the crisp value is lower than 0.
        v.  Calculate the F1 metric.
        vi.  Optimize the parameters of linear functions with CMA-ES, defining the F1 metric as the fitness function.
        vii.  Evaluate the model on the validation set when the fitness function is optimized or the maximum number of evaluations passes. Otherwise return to (iii).
    b.  Choose a set of hyperparameters which maximize the mean value of the F1 metric over the validation sets.

In order to evaluate the model for each dataset, we conduct the following procedure:

(1)  Choose the best hyperparameters set for a dataset.
(2)  Optimize the fitness function on the training set.
(3)  Choose the best parameters of linear functions.
(4)  Evaluate the model on a test set.

Repeat the procedure *n* times (*n* = 10) to minimize CMA-ES possible local extrema solutions and choose the best result.

## 4. Binary Classification Results

In Table 8 the best binary classification results achieved for the best hyperparameters setup are presented.

**Table 8.** Classification results.

| Dataset | F1 Score (%) | Accuracy (%) | Sensitivity (%) | Hyperparameters | ROS | PCA |
|---|---|---|---|---|---|---|
| Blood Transfusion | 56.6 | 69.3 | 83.3 | <11 Gaussian, equal, mean, 0.75> | Yes | No |
| Breast Cancer Data | **97.6** | **98.2** | **95.2** | <3 Gaussian, equal, center, 0.5> | Yes | Yes |
| Breast Cancer Wisconsin | **96.0** | **97.1** | **100.0** | <3 Gaussian, progressive, mean, 0.85> | Yes | Yes |
| Data Banknote Authentication | **100.0** | **100.0** | **100.0** | <3 Gaussian, equal, mean, 0.65> | Yes | No |
| Haberman | 40.0 | 66.1 | 43.8 | <9 Gaussian, equal, mean, 0.55> | No | No |
| Heart | **85.2** | **85.2** | **92.9** | <3, progressive, mean, 0.6> | Yes | Yes |
| HTRU 2 | **87.8** | **97.8** | **85.4** | <9 Gaussian, equal, center, 0.5> | No | Yes |
| Immunotherapy | 66.7 | 83.3 | 75.0 | <9 Gaussian, equal, mean, 0.75> | Yes | No |
| Indian Liver Patient | 57.4 | 58.1 | 97.1 | <11, Gaussian, equal, center, 0.8> | Yes | Yes |
| Ionosphere | **89.4** | **93.0** | **84.0** | <7 Gaussian, equal, mean, 0.55> | No | Yes |
| Parkinson | 80.0 | 89.7 | 80.0 | <5 Gaussian, progressive, mean, 0.8> | Yes | No |
| Pima Indians Diabetes | 66.2 | 68.8 | 87.0 | <11 Gaussian, equal, center, 0.9> | Yes | Yes |
| Vertebral | **87.1** | **82.3** | **88.1** | <7 Gaussian, progressive, mean, 0.75> | Yes | No |

In Table 9, we present the best classification results achieved in comparison with our previous research [22] and other classifiers as well.

**Table 9.** Comparison with other methods, concerning the F1 score measure.

| Dataset | The Presented Approach (Using the Takagi–Sugeno Model) (%) | Our Previous Approach (Using the Mamdani Model) (%) | Other Classifiers (%) |
|---|---|---|---|
| Breast Cancer Data | 97.6 | 91.2 | Immune-inspired semi-supervised Algorithm, introduced in [44]: 97.3 |
| Breast Cancer Wisconsin | 96.0 | 95.7 | Extreme Learning Machine Neural Networks, introduced in [45]: 97.8 Immune-inspired semi-supervised algorithm, introduced in [44]: 96.5 Support vector machines combined with Feature Selection, introduced in [46]: 99.7 |
| Data Banknote Authentication | 100.0 | 99.3 | Deep Neural Network with PCA and LDA, introduced in [47]: 99 Decision tree approach, introduced in [48]: 99.4 Random Forest approach, introduced in [49]: 94.8 Neural Network-Genetic Algorithm, introduced in [50]: 100 |
| HTRU 2 | 87.8 | 89.0 | Classical classifiers: C4.5: 74; MLP: 75.2; NB: 69.2; SVM: 78.9 GH-VFDT Algorithm, introduced in [51]: 86.2 A hybrid ensemble method, introduced in [52]: 91.8 (with respect to a voting threshold parameter) |
| Ionosphere | 89.4 | 88.8 | Clustered Bayesian classification, 88.5 [53] |

## 5. Discussion

This research is a continuation of a previous one, published in [22]. Both papers present our newly introduced definition of a fuzzy information system, which has the following advantages:

- The information function values are interpreted as fuzzy sets, labeled with corresponding linguistic variables. This gives the possibility to generalize information—we do not consider numerical values for pairs (object, attribute), but general descriptions such as 'small', 'medium', and 'high'.
- The decision table used is generated in an automatic manner for a considered data set, as the value 'medium' is assumed as the Gaussian distribution of the data for each attribute. Then, the sets 'low' and 'high' are easy to be defined using the 'medium' membership function. Next, a corresponding label (identifying the corresponding fuzzy set) is given for a pair (object, attribute), by using the maximum membership value.

Additionally, we propose an easy transformation in order to change the applied type-1 fuzzy sets into type-2 fuzzy sets.

Therefore, by defining such a decision table, the rule induction procedure described in [38,39] is applied and next, type-2 fuzzy rules are derived. What is more, different fuzzification strategies are possible (see Table 4).

In this research, the goal was to incorporate our fuzzy information system concept with the Takagi–Sugeno model, as it does not require the model outputs to be defined as fuzzy sets. Such an achievement would create new possibilities, using the advantages of the Takagi–Sugeno model. To incorporate the model, we had to optimize the model parameters with respect to the corresponding benchmark data. We have applied the CMA-ES (Covariance Matrix Adaptation Evolutionary Strategy) for this purpose. We conducted experiments with the same binary classification problems and the same fuzzification experiments, as presented in [22], in order to compare the results.

While performing experiments, we have discovered further promising features and potential disadvantages of the method proposed. The classification process is able to fit training data, but sometimes has issues with generalization. Applying PCA allows us to minimize this issue, making the method proposed more robust. Additionally, performing optimization with an evolutionary strategy allows the model to find very well-fitting decision boundaries. Therefore, one might achieve even better performance by applying additional regularization methods.

To solve the problem of imbalanced data, Random Oversampling was used, which was not beneficial in a few cases. If the imbalance is very high, ROS may produce too much redundancy; therefore, the information system may produce biased rules, ignoring important information contained in the data. In such a scenario, other sampling techniques should be applied instead.

Regarding the attribute fuzzification procedure, only Gaussians were assumed as membership functions. We suppose, there is room for improvement, adjusting membership functions to the attributes' characteristics. Additionally, not only the first-order Takagi–Sugeno model could be applied.

Despite such constraints, good classification results for several datasets were achieved. We believe that the application of the Takagi–Sugeno model in our approach will support the solving of multi-class classification problems. This is because, in the model used, there is no need to define decision classes, contrary to the Mamdani fuzzy model.

Summarizing, a new classification concept was proposed in this research with the following main advantages:

- Defining a decision table with fuzzy values. The fuzzification is provided in an automatic manner directly from a data set.
- Using the rule induction method based on the information system concept, which has a solid mathematical background. Each rule is related to a corresponding class, regarding the classification problem considered.
- Transformation of the induced rules into type-2 fuzzy rules.
- Application of the Takagi–Sugeno model in the classification process. Therefore, there is no need to define the fuzzy rule consequents as fuzzy sets.

## 6. Conclusions

In this research, a novel classification method, which incorporates our previously introduced concept of a fuzzy information system with the first-order Takagi–Sugeno model, was introduced. A fuzzification procedure with membership function optimization and a corresponding rule induction of type-2 fuzzy rules were proposed as well. The research was a continuation of our previous approach. Here, the aim was to replace the fuzzy model previously applied with the Takagi–Sugeno model. The model parameters optimization was conducted with the CMA-ES evolutionary strategy with respect to the classification problem considered. The achieved classification results proved that the Takagi–Sugeno-type inference is fully implementable with our fuzzy type-2 rule induction concept. Therefore, there is no need to define the fuzzy rule consequents as fuzzy sets, which simplify the whole classification method and potentially support the solving of multi-class classification problems. Nevertheless, some improvements could be applied, such as better feature selection, detection of outliers, or data augmentations in order to increase the classification results. Limitations of the method proposed might occur if the

generalizations applied in the decision table are too high. This means that if there is a low number of fuzzy sets and attributes, the rules induced will not differentiate the decision classes appropriately. Additionally, categorical variables are not appropriate due to the required data fuzzification.

As further research, we consider the possibility to use representation learning in order to generate attribute values for a considered problem and to fuzzify them. Therefore, we intend to involve the deep learning phase in the method proposed. We consider the concept proposed as a general approach, which is applicable to real-world classification problems, meant for data containing vague and incomplete information.

**Author Contributions:** Conceptualization, M.T.; Methodology, M.T.; Software, A.B.C. and A.R.C.; Validation, A.B.C. and A.R.C.; Formal analysis, M.T.; Investigation, M.T., A.B.C. and A.R.C.; Data curation, A.B.C. and A.R.C.; Writing—original draft, M.T.; Writing—review & editing, A.B.C. and A.R.C.; Supervision, M.T. All authors have read and agreed to the published version of the manuscript.

**Funding:** This work was supported by the statutory funds of the Department of Artificial Intelligence, Wroclaw University of Science and Technology.

**Institutional Review Board Statement:** Not applicable.

**Informed Consent Statement:** Not applicable.

**Data Availability Statement:** The presented experiments were conducted on publicly available benchmark data.

**Acknowledgments:** All calculations were performed using computational resources provided by WCSS (Wroclaw Centre for Networking and Supercomputing). Additionally, we wish to acknowledge the help provided by Konrad Karanowski, Aleksy Walczak, Mateusz Grzesiuk, Jan Wielgus, Artur Zawisza and Piotr Majchrowski, for collaboration in implementation of software library dedicated to fuzzy techniques, which we have applied in our experiments.

**Conflicts of Interest:** The authors declare no conflict of interest.

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
