# Peer review of "A Novel Classification Method Using the Takagi–Sugeno Model and a Type-2 Fuzzy Rule Induction Approach"

_applsci, doi:10.3390/app13095279_

Round 1

Reviewer 1 Report

The manuscript "Classification with Fuzzification Optimization using the Takagi-Sugeno Model and Type-2 Fuzzy Sets" deal with classification data by using a type2 fuzzy logic approach with an evolutionary algorithm. 

The abstract and Conclusion need to be rewritten by IMRAD structure. 

And the authors should clarify the main scientific goal of their research. All approaches are well-known. If they use this approach for untypical tasks - it could be also the goal of the paper. 

The introduction section contains a short review of the field of classification methods and is well-written.

The second section provides a description of the basic principles and procedures of type-1 and type-2 fuzzy logic approaches and explains the data.

The third section deals with the membership functions of the type-2 fuzzy logic approach and describes the well-known evolution algorithm. 

In the fourth section, the authors give the results of binary classification with the comparison of their first approach.

What is the main scientific goal of this research? Authors should clarify it and after this gives a good extensive analysis of the results obtained.

Reviewer 2 Report

This paper employs the Takagi-Sugeno Model and Type-2 Fuzzy Sets and uses evolutionary strategy to determine the best fuzzy classifier for presented benchmark data. The authors should be commended for proposing a novel and interesting approach to classification. In my opinion, minor revisions are necessary and the following comments improves the paper's quality.

11) I suggest that the authors change the title. The term “Classification with Fuzzification Optimization” is vague and somehow misleading.

T2) The following sentence on page 1 needs to be revised.: “The introduced by Pawlak [9-11] information systems and rough sets, are often used to represent knowledge”

33) The introduction is short and needs to be expanded in order to make the paper more informative for readers. Additionally, there are many other relevant articles that have used the same benchmark data that are not covered in this article.

44) In section 2.2, the membership symbol should be written consistently and with subscripts

55) You should define all parameters in the equations. There are a few occasions when it hasn't been done.

66) The illustration provided by table 2 and figure 2 is not clear. You should provide the necessary explanations.

77) The conclusion is short and needs to be expanded

Reviewer 3 Report

The reviewed manuscript follows-up the previous research of the authors, wherein the authors establish a classification procedure on the basis of interval type-2 fuzzy rules induction. In their current research, to justify the approach designed and comparative analysis conducted in the classification process, the authors implement the Takagi-Sugeno inference model. The model parameters are optimized by applying specific evolutionary algorithm. 

Therefore, this topic addressed is very interesting and adequate methods of investigation are applied. Furthermore, the topical research is supported by a wide array of experimental and graphical outputs, i.e., thorough description of a modeled procedure, tables, charts. The structure of the manuscript is well compiled and discussion of individual findings is comprehensive.

However, multiple minor (major) deficiencies have been detected in the text (and they need to be revised prior publishing the paper). See them as follows:

Generel remarks:

1.) Abstract must be extended, wherein the manuscript structure needs to be briefly outlined and the novelty of the research conducted is to be emphasized. Moreover, I highly recommend the authors conveying the subject (or field of study / area of investigation) to which the performed research study "Fuzzification Optimization using the Takagi-Sugeno Model and Type-2 Fuzzy Sets" was based (connected). Or, truly was the whole research just generally-conceived? In my opinion, the field (subject) of research conducted should be emphasized in the paper.

2.) Prior to "Material and methods" section, in an effort to enhance the scientific soundness of the paper, the authors are highly required to elaborate a self-standing part entitled "Literature review", in which a number of topic-related and WoS (Scopus) indexed references must be incorporated. Please take into account even the ensuing citations:

a) (2020). Possible Application of Solver Optimization Module for Solving Single-circuit Transport Problems. LOGI – Scientific Journal on Transport and Logistics, 11(1), pp. 78-87. DOI: 10.2478/logi-2020-0008;

b) (2022). Navigation of three cooperative object-transportation robots using a multistage evolutionary fuzzy control approach. IEEE Transactions on Cybernetics, 52(5), pp. 3606-3619. DOI: 10.1109/TCYB.2020.3015960;

c) (2019). Application of the Operational Research Method to Determine the Optimum Transport Collection Cycle of Municipal Waste in a Predesignated Urban Area. Sustainability, 11(8), Article no: 2275. DOI: 10.3390/su11082275;

d) (2022). A coordinated control strategy for frequency regulation in hybrid shipboard power system using novel salp swarm algorithm tuned fractional controller. International Journal of Ambient Energy, 43(1), pp. 5638-5653. DOI: 10.1080/01430750.2021.1973558;

e) (2022) Algorithm for Creating Optimized Green Corridor for Emergency Vehicles with Minimum Possible Disturbance in Traffic. LOGI – Scientific Journal on Transport and Logistics, 13 (1), pp. 84-95. DOI: 10.2478/logi-2022-0008;

f) (2022). Modelling Distribution Routes in City Logistics by Applying Operations Research Methods. Promet - Traffic&Transportation, 34(5), pp. 739-754. DOI: 10.7307/ptt.v34i5.4103;

g) (2020). Fuzzy logic controller to control the active air suspension. The Archives of Automotive Engineering – Archiwum Motoryzacji, 89(3), pp. 75-86. DOI: 10.14669/AM.VOL89.ART6;

h) (2021). Fuzzy and evolutionary algorithms for transport logistics under uncertainty. Advances in Intelligent Systems and Computing, 1197, pp. 1456-1463; In: 2021 International Conference on Intelligent and Fuzzy Systems, INFUS 2020, Istanbul, Turkey, 21-23 July 2020. DOI: 10.1007/978-3-030-51156-2_169.

and so on.

3. Last but not least, "Conclusions" part needs to be considerably extended. Please try to add even some limitations of the research conducted as well as several recommendations for further research in an issue addressed.

Specific remarks:

1.) All the equations need to be referred (numbered) even in the text prior to the equations itselves.

2.) I suggest rephrasing all the expressions (formulations) in the first person plural; for instance: "in which we introduced a classification procedure...", by expressions in passive, like this: "a classification procedure was introduced...". Or, "Our experimental procedure pipeline was defined as follows...", by "The experimental procedure pipeline in this research was defined as follows...". It looks more appropriate and adequate for high-quality scientific journals.

Round 2

Reviewer 1 Report

The manuscript can be accepted.

Reviewer 3 Report

As a matter of fact, the authors have disregarded several of my previous remarks and recommendations, therefore, I do not change my previous review report decision.

1.) I still urge the authors to elaborate a self-standing manuscript part entitled "Literature review", in which a number of topic-related and WoS (Scopus) indexed references, which entails a basic standard for a high-quality scientific papers.

2.) "Conclusions" part is still compiled poorly and is notably short. Hence, it needs to be considerably extended. Please try to add even some limitations of the research conducted. Apart from that, recommendations for further research are sadly poor, thus must be significantly expanded and specified.

3.) All the expressions (formulations) in the first person plural; for instance: "In both cases to induce fuzzy rules, we used a new concept of a fuzzy information system...", or "A key point in our ap-proach is the generalization of the fuzzy information systems’ attribute information...", have not been rephrased/replaced by expressions in passive.

4.) In addition, I suggest meticulous and profound text proofreading, as multiple English grammar errors occur (such as wrong word order and so on).
